# Middle East Respiratory Syndrome Coronavirus (MERS-CoV) in Dromedary Camels in Africa and Middle East

**DOI:** 10.3390/v11080717

**Published:** 2019-08-05

**Authors:** Ahmed Kandeil, Mokhtar Gomaa, Ahmed Nageh, Mahmoud M. Shehata, Ahmed E. Kayed, Jamal S. M. Sabir, Awatef Abiadh, Jamel Jrijer, Zuhair Amr, Mounir Abi Said, Denis K. Byarugaba, Fred Wabwire-Mangen, Titus Tugume, Nadira S. Mohamed, Roba Attar, Sabah M. Hassan, Sabah Abdulaziz Linjawi, Yassmin Moatassim, Omnia Kutkat, Sara Mahmoud, Ola Bagato, Noura M. Abo Shama, Rabeh El-Shesheny, Ahmed Mostafa, Ranawaka A. P. M. Perera, Daniel K. W. Chu, Nagla Hassan, Basma Elsokary, Ahmed Saad, Heba Sobhy, Ihab El Masry, Pamela P. McKenzie, Richard J. Webby, Malik Peiris, Yilma J. Makonnen, Mohamed A. Ali, Ghazi Kayali

**Affiliations:** 1Center of Scientific Excellence for Influenza Virus, Environmental Research Division, National Research Centre, Giza 12622, Egypt; 2Center of excellence in Bionanoscience Research, King Abdulaziz University, Jeddah 80203, Saudi Arabia; 3Biotechnology Research Group, Department of Biological Sciences, Faculty of Science, King Abdulaziz University, Jeddah 80203, Saudi Arabia; 4Nature Link, Sfax 3000, Tunisia; 5Department of Biology, Jordan University of Science and Technology, Irbid 22110, Jordan; 6Department of Life and Earth Sciences, Faculty of Sciences II, Lebanese University, Al Fanar 90656, Lebanon; 7Department of Epidemiology and Biostatistics, School of Medicine, Makerere University, Kampala 7062, Uganda; 8Department of Genebank and Genetic Sequence, Forensic DNA Research and Training Center, Al-Nahrain University, Baghdad 10072, Iraq; 9Department of Biological Sciences, King Abdulaziz University, Jeddah 80203, Saudi Arabia; 10Princess Doctor Najla Saud Al-Saud Distinguished Research Center for Biotechnology, Jeddah 22252, Saudi Arabia; 11Department of Genetics, Faculty of Agriculture, Ain Shams University, Cairo 11241, Egypt; 12King Fahad Medical Center, King Abdulaziz University, Jeddah 80203, Saudi Arabia; 13St. Jude Children’s Research Hospital, 262 Danny Thomas Place, Memphis, TN 38105, USA; 14School of Public Health, The University of Hong Kong, 7 Sassoon Rd, Hong Kong, China; 15General Organizations of Veterinary Services, Ministry of Agriculture and Land Reclamation, Nadi Saed St. 1, Dokki, Giza 12618, Egypt; 16Food and Agriculture Organization of the United Nations, Emergency Center for Transboundary, Animal Diseases, Dokki, Giza 12611, Egypt; 17Animal Health Services (AGAH), Emergency Centre for Transboundary Animal Diseases (ECTAD), Dokki, Giza 12611, Egypt; 18Human Link, Hazmieh 1109, Lebanon; 19Department of Epidemiology, Human Genetics, and Environmental Sciences, University of Texas, Houston, TX 77030, USA

**Keywords:** MERS coronavirus, surveillance, virus infection, epidemiology, virus transmission

## Abstract

Dromedary camels are the natural reservoirs of the Middle East respiratory syndrome coronavirus (MERS-CoV). Camels are mostly bred in East African countries then exported into Africa and Middle East for consumption. To understand the distribution of MERS-CoV among camels in North Africa and the Middle East, we conducted surveillance in Egypt, Senegal, Tunisia, Uganda, Jordan, Saudi Arabia, and Iraq. We also performed longitudinal studies of three camel herds in Egypt and Jordan to elucidate MERS-CoV infection and transmission. Between 2016 and 2018, a total of 4027 nasal swabs and 3267 serum samples were collected from all countries. Real- time PCR revealed that MERS-CoV RNA was detected in nasal swab samples from Egypt, Senegal, Tunisia, and Saudi Arabia. Microneutralization assay showed that antibodies were detected in all countries. Positive PCR samples were partially sequenced, and a phylogenetic tree was built. The tree suggested that all sequences are of clade C and sequences from camels in Egypt formed a separate group from previously published sequences. Longitudinal studies showed high seroprevalence in adult camels. These results indicate the widespread distribution of the virus in camels. A systematic active surveillance and longitudinal studies for MERS-CoV are needed to understand the epidemiology of the disease and dynamics of viral infection.

## 1. Introduction

Middle East respiratory syndrome coronavirus (MERS-CoV) was first discovered in a patient suffering from acute pneumonia and renal failure on June 13, 2012 in the Kingdom of Saudi Arabia (KSA) [1]. Among 2434 MERS-CoV human infection cases reported to the World Health Organization (WHO) from 27 countries as of 22 May, 2019, 876 deaths have been reported with case fatality rate of 35.9% [2]. Serological and molecular studies suggested that the main zoonotic source for MERS-CoV is dromedary camels, but the main origin of the virus is still unclear [3,4,5]. Epidemiological and viral sequence data suggest that camels are the main source for virus transmission to humans [6,7]. Human to human transmission was recorded in clusters and outbreaks from the Arabian Peninsula and South Korea [8,9,10]. 

Dromedary camels are part of the heritage of millions of people in Egypt and other Middle Eastern and African countries. Camel meat is an important source of nutrition for several communities. Most dromedary camels traded in the Middle East are bred in Eastern African countries, primarily in Ethiopia, Sudan, Somalia, and Kenya [11]. Dromedaries from African countries (Egypt, Ethiopia, Kenya, Senegal, Burkina Faso, Nigeria, Sudan, and Tunisia) and Arabian Peninsula (Jordan, Oman, Qatar, KSA, and United Arab Emirates) have high seropositive rates of MERS-CoV [12,13,14,15,16]. A retrospective serological study conducted on archived sera (1983 to 1997) unveiled the presence of neutralizing antibodies in camels [17] suggesting long-term circulation of MERS-CoV among camels, but lack of surveillance and absence of knowledge of the virus delayed its detection.

There is no zoonotic MERS disease reported in Africa. This may be due to limited epidemiological surveillance for MERS-CoV in Africa, differences in genetic characteristics of circulating MERS-CoV in sub-Saharan Africa and Middle Eastern countries, or other human host factors. Viral genetic differences in detected viruses from different origin may be the main well-known relevant factor of zoonotic potential.

The current recommended control strategies for MERS-CoV infection in camels are regular active surveillance, control of camel movement in the infected areas, use of personal protective equipment during handling of camels, increasing awareness about the virus and the risks of exposure to unpasteurized camel milk, raw meat, viscera, and urine [18].

This paucity of data and absence of prospective surveillance in animals contribute to our imperfect understanding of the epidemiology and risk factors associated with zoonotic MERS. There is limited data on the prevalence of MERS-CoV in different countries. We designed an active surveillance program to study MERS-CoV circulation among dromedaries in Egypt, Senegal, Tunisia, Uganda, Jordan, KSA, and Iraq. We also carried out a longitudinal study of three camel herds in Egypt and Jordan to elucidate MERS-CoV infection and transmission.

## 2. Materials and Methods

### 2.1. Sampling and Locations

Samples were collected from dromedary camels from Egypt, Senegal, Uganda, Jordan, Iraq, Tunisia, and KSA (Figure 1). A total of 2230 nasal swabs and 2033 serum samples were collected from camels in Egypt between April 2016 and March 2018 from eight governorates including different sampling sites (164 from quarantines, 286 from live animal markets, 649 from slaughterhouses, 187 from free-roaming herds, and 944 from farms). The majority of camels were imported from Sudan and sampled within no more than 10 days of importation. Two local farms breeding herds in Esna (South of Egypt, 49 camels) and Matrouh (Northwest coastal region, 65 camels) were longitudinally sampled. The same animals were resampled when possible. In Senegal, a total of 127 nasal swabs and 198 serum samples of camels were collected from seven areas (Gandon, Ndaye, Gantour, Ndoye, Rao, Tongon, and Toug) in the Saint-Louis region located in the northwest of the country during August and September 2017. All camels were pastoralist domestic. In Tunisia, a total of 1170 nasal swabs and 782 serum samples were collected from adult camels at seven markets (Douz, Ghlissia, Gollaa, Zaafrana, Ksar Ghilane, Ghidma, and Jamnah) during December 2015 to January 2018. In Uganda, 250 dromedary pastoralist domestic camels were sampled from February to March 2017, nasal and serum sample were obtained from each camel. Camels were collected from Moroto and Amudat districts located at Northern East Uganda. In Iraq, we collected 32 serum samples, and 26 nasal swabs during January 2017 from local camels at Wasit and Muthana governorates. Furthermore, 224 nasal swabs and 222 blood samples were collected from farm camels in the Western region of KSA. A herd of farm dromedary camels in Jordan was longitudinally sampled for nasal and serum samples between November 2015 and October 2016.

### 2.2. Serological Testing

Serum micro-neutralization assay was conducted as described by Perera et al. (2013) [13], using Vero-E6 cell monolayers. Briefly, serial two-fold dilutions of 200 µL heat-inactivated sera (56 °C for 30 min) were made, starting with a dilution of 1:10. The serum dilutions were mixed with equal volumes of 200 tissue culture infectious dose (TCID_50_) of dromedary MERS-CoV Egypt NRCE-HKU270. After 1h of incubation at 37 °C, 35 µL of the virus–serum mixture was added in quadruplicate to Vero-E6 cell monolayers in 96-well microtiter plates. After 1 h of adsorption, an additional 150 µL of culture medium was added to each well. The plates were then incubated for three more days at 37 °C in 5% CO_2_ in a humidified incubator. A virus back-titration was performed without immune serum to assess input virus dose. Cytopathic effect (CPE) was read at three days post infection. The highest serum dilution that completely protected the cells from CPE in at least half of the wells was considered as the neutralizing antibody titer and was estimated using the Reed–Muench method. Positive cut off points were set at values greater or equal to 1:20 serum dilution points.

### 2.3. Molecular Testing

Real-time reverse transcription polymerase chain reaction (rtRT-PCR) targeting upstream of E gene of MERS-CoV was used for screening [19]. Confirmation was made using assays targeting the Open Reading Frame (ORF) 1a or N gene based on WHO recommendations for MERS-CoV diagnosis [20]. A partial 701 bp fragment of the spike gene containing the RBD was amplified using pre-RBD-MERS-F (GAATCTGGAGTTTATTCAGTTTCGT) and pre-RBD-MERS-R (ACGGCCCGA AACACCATAG) primers in the first round using one step RT-PCR kit (QIAGEN, Germany). The PCR products were then subjected to a second PCR round with RBD-MERS-F: (CGAAGCAAAACCTTCTGGCT) and RBD-MERS-R: (ATATTCCACGCA ATTGCCTA) primers and using Phusion High Fidelity PCR Master Mix Kit (Thermo Scientific, Waltham, MA, USA). The final PCR product was gel purified then sequenced with the same primers at the Macrogen sequencing facility (Macrogen, Seoul, South Korea). The phylogenetic tree was constructed using MEGA6 program by applying the neighbor-joining method with Kimura’s two-parameter distance model and 1000 bootstrap replicates.

### 2.4. Statistical Analysis

All statistical analyses were performed using SPSS version 16 for windows. The association between MERS-CoV prevalence in camels and the study variables (sampling site, origin, age, and sex) were analyzed by Pearson Chi-square test of independence. Statistical significance was considered at *p*-value less than 0.05.

### 2.5. Ethics Approval

This study was carried out in accordance with the principles of the Animal Welfare Act of the United States of America. The protocol was approved by the Institutional Animal Care and Use Committee at St. Jude Children’s Research Hospital. Also, the ethics committee of the National Research Centre, Egypt, approved the animal sampling protocol (Approval code: 17-070).

## 3. Results

The overall seroprevalence of MERS-CoV antibodies in all collected sera (3821 samples) from the seven countries (Egypt, Uganda, Senegal, Tunisia, Jordan, KSA, and Iraq) as illustrated in Figure 1 was 73.7% with seroprevalence of 30.3% in juvenile (<2 years old) and 82.6% in adult camels (>2 years old) (Figure 2). The prevalence rate of MERS-CoV RNA in 4331 nasal swabs collected from the seven countries was 4.7%, with virus detection rates of 5.5% and 1.4% in adult and juvenile, respectively. Male camels had a significantly higher detection rate of MERS-CoV than female camels (8% and 1.2%, respectively, *p*-value < 0.001) (Figure 2). No significant difference in the seroprevalence of MERS-CoV antibodies in male and female camels was detected (76.5% and 71.5%, respectively).

### 3.1. Egypt

Of the 2033 serum samples tested, 1401 (68.9%) were positive (Figure 1). Seroprevalence rates differed significantly by sampling collection site, governorate, sex, age, and origin of animals (*p*-value < 0.0001) (Table 1). Camels sampled at slaughterhouse had the highest seroprevalence (89.3%), while camels from farms had the lowest seroprevalence (51.2%, *p*-value < 0.0001).

Among eight governorates of Egypt, Qalubiya showed the highest seroprevalence rate (93.5%), followed by Giza and Aswan (88.2%). Juvenile camels had a significantly lower seroprevalence than adult camels (16.1% and 84% respectively, *p*-value < 0.0001). Males tended to have a higher seropositivity rate than females (75.4% and 55% respectively, *p*-value < 0.0001). Imported camels had a higher seropositivity rate than local camels (86.5% and 54.2% respectively, *p*-value < 0.0001).

MERS-CoV RNA detection rate from nasal swabs of camels in Egypt was 8.2% overall (Figure 1) (183 positive samples out of 2230 tested; Table 1). The virus’ RNA was detected in all sampling sites with the highest virus detection rate in slaughterhouse (20%) followed by live animal markets (4.1%), farms (3.4%), free herds (1.6%), and quarantines (1.2%). The highest detection rate of virus RNA (21.5%) was in Cairo followed by Giza (17.2%) and Qalubiya (13.5%). However, adult camels tested positive more than juveniles (*p*-value < 0.0001). MERS-CoV was detected more in imported camels than local ones (13.3% and 3.1% respectively, *p*-value < 0.0001).

### 3.2. Senegal

The seroprevalence of MERS-CoV in 198 serum samples collected from camels in Senegal was 65.1% while virus detection rate was 5.5%. Seroprevalence rates differed significantly by sampling site and age. Adult camels had a higher seropositivity rate (68%) than juvenile (29.4%) (*p*-value < 0.01). There was no statistical difference in seroprevalence between female and male in camels of Senegal (*p*-value > 0.05).

The virus’ RNA was detected in four villages (Gandon, Ndaye, Gantour, and Rao). There was no significant difference in the detection rate of MERS-CoV in camels of Senegal based on age or sex (*p*-value > 0.05).

### 3.3. Uganda

Camels from both Morto and Amudat at the Northern East region had comparable (*p*-value > 0.05) levels of seroprevalence (66.3% and 64.9%, respectively). There was no statistically significant difference in seroprevalence between adult and juvenile camels of Uganda. The seroprevalence was significantly higher in male (78%) than in female camels (61.3%) (Table 1) (*p*-value < 0.05). No virus was detected in any nasal swab from camels in Uganda.

### 3.4. Tunisia

Of the 782 serum samples from camels, 683 (87.3%) were seropositive for MERS-CoV. Juvenile camels had significantly higher seroprevalence (100%) than adult camels (86.8%) (*p*-value < 0.02). Of the 1170 nasal samples from camels, nine (0.76%) were positive for MERS-CoV. Of the nine positive camels, one was located at Ghlissia, and eight at Ksar Ghilane (Table 1). The number of confirmed PCR-positive MERS-CoV cases was significantly higher in male than female camels (*p*-value < 0.001). There was no statistical difference in the detected MERS-CoV between adult and juvenile camels in Tunisia (*p*-value > 0.05).

### 3.5. KSA

Of the 222 serum samples from camels, 181 (81.5%) were seropositive. There was no statistical difference by age, sex, or sampling location of serum samples (*p*-value > 0.05). Of the 224 nasal swabs, MERS-CoV RNA was detected in seven samples (3.1%). There was no statistical difference in the detected MERS-CoV by age or sex (*p*-value > 0.05). Among nine regions where camels were sampled, the virus was detected in Alkhomra, Dahban, and Alad.

### 3.6. Jordan

The virus was not detected in a total of 304 nasal swabs collected from camels in Jordan. The seroprevalence rate of MERS-CoV in 304 serum samples was 81%. The seroprevalence was significantly higher in female (82.1%) than in male camels (28.5%) (Table 1) (*p*-value < 0.003). Adult camels had significantly higher seroprevalence (92.3%) than young camels (50%) (*p*-value < 0.0001).

### 3.7. Iraq

The virus was not detected in a total of 26 nasal swabs collected from camels in Iraq. The seroprevalence rate in 32 serum samples was 43.7%. There was no statistical difference by age, sex, or sampling location of sera and nasal swabs collected from Iraq.

### 3.8. Sequence Analysis

A phylogenetic tree was constructed based on 11 partial spike nucleotide sequences of RBD obtained from strongly positive samples (two from KSA, three from Senegal, and six from Egypt) and representative published sequences from different countries. The generated phylogenetic tree for partial spike nucleotide sequences of RBD had a topology similar to that of whole genome virus. The tree suggested that sequences from camels in Egypt formed a separate group from previously published sequences of MERS-CoV. The new Egyptian sequences clustered together with African MERS-CoV detected in Ethiopia, Burkina Faso, Morocco, and Nigeria in clade C. In addition, two sequences obtained from camels in KSA were distinct from previously detected viruses in Saudi Arabia and closely related to clade C. The three sequences obtained from Senegal were closely related to cluster B circulating in Asia (Figure 3).

### 3.9. Longitudinal Studies

Camels from a herd of 65 camels (18 male and 47 female) including 31 adult and 34 juveniles residing on a breeding farm at Matrouh governorate in Egypt were sampled monthly from May 2016 to February 2017 (except October). The seroprevalence rate of MERS-CoV in camels ranged from 29% to 46% as illustrated in Figure 4A. MERS-CoV antibody was not detected in juvenile camels. The overall geometric mean titer of MERS-CoV antibodies within the herd was 5.3 log_2_ (Figure 4B).

MERS-CoV RNA was detected in nasal swabs from five dromedaries in May 2016 (7.69%). At the time of sampling, of the five infected camels (two males and three females), two camels were seronegative while the three remaining camels had antibody titers ranging from 160 to 320. MERS-CoV RNA was detected only in one nasal swab collected in June 2016 from a seronegative adult female camel in which infection was not previously detected. This camel became seropositive when sampled in July and remained so till the end of the study. Three camels were found positive for MERS-CoV infection in July 2016 (two seronegative and did not sero-convert, and one had 320 antibody titer and continued to be seropositive). Two camels positive for MERS-CoV infection in samples collected in August (one was seronegative and never sero-converted and the other had a titer of 320 and continued to be seropositive). The virus was not detected in samples collected in September, but it was detected in only one swab of samples collected in November that had 80 antibody titer and become seronegative in subsequent samplings. No virus was detected in collected samples during December to February 2017. Steady-state of MERS-CoV antibody titer was observed during longitudinal study at Matrouh (Figure 4B).

A herd of 49 adult camels imported from Sudan and locally residing in a breeding farm at Esna in Egypt, were sampled four times from 20 April 2016 to 30 May 2016. The seroprevalence rate of MERS-CoV in camels was 97.9% at the baseline with 8.6 log_2_ geometric mean titer. Seroprevalence rate of MERS-CoV at the first follow up was 7.6 log_2_ geometric mean titer as illustrated in Figure 4A. At the second follow up, the seroprevalence rate of MERS-CoV decreased to 5 log_2_ geometric mean titer then slightly increased at the third follow up to 5.1 log_2_ geometric mean titer. The virus was detected in 13 camels at the third follow up [2 (<10), 2 (20), 8 (40), 1 (80)].

Viral infection was not detected in any of the camels sampled from Jordan’s herd (Figure 4A). All the camels that were sampled had MERS-CoV antibodies ranging from 6.8 to 8 log_2_, thus indicating past infection (Figure 4B).

## 4. Discussion

This study confirms the extensive circulation of MERS-CoV within the dromedary camel population in Egypt and Tunisia (North Africa), Senegal (West Africa), Uganda (East Africa), Jordan (Middle East), and KSA (Arabian Peninsula). The virus was detected in nasal swabs of seropositive dromedary camels indicating reinfection of seropositive animals. A previous study showed that the lack of correlation between MERS-CoV shedding and neutralizing antibodies was due to the ability of the virus to re-infect camels having MERS-CoV antibodies [21]. The tested camels in seven countries had MERS-CoV neutralizing antibodies and virus RNA was detected in Egypt, Senegal, KSA, and Tunisia. None of the infected camels in the four countries showed any clinical signs related to MERS infection as it is known that infected camels rarely show any signs of infection [22].

The main movement flow of dromedary camels in Africa is in the south-north direction rather than east–west movements [23]. Egypt annually imports large numbers of live dromedary camels from Sudan, Ethiopia, and Somalia to cover a part of its animal protein need. Some Sudanese camels are trekked to Egypt via the traditional route “Fourties road”, but most of camels are transported from Sudan to Egypt by trucks to live animal markets (Draw and Birqash) The Birqash camel market in Giza governorate in Egypt is the largest market in Africa. Camels from horn of African countries are shipped to Egypt via the Red Sea then transported from ports to a quarantine station then directly to markets by trucks. Most of imported camels are slaughtered and a few of them are kept for fattening for several weeks before being put up for sale again. During our previous cross-sectional surveillance study in Egypt, results showed that 84.5% of tested camels had MERS-CoV neutralizing antibodies in their sera while 3.8% of camels were positive for MERS-CoV [16]. In the present study, 68.9% of camels in Egypt were seropositive for MERS-CoV while virus genetic materials were detected in 8.2% of investigated camels. The prevalence of MERS-CoV varied significantly by sampling collection site, governorate, sex, age, and origin of animals. Similar to our previous studies, analysis of the results based on origin of camels in Egypt indicated that imported camels had higher seroprevalence rate and rtRT-PCR positive cases of MERS-CoV than resident camels [16]. It appears that the detection rates increase as camels move through the value-chain and reaches its highest at abattoirs. It is likely that the infection is amplified as animals are brought to close proximities in markets and abattoirs. This study showed that examined adult camels in Egypt had higher seroprevalence (84%) and viral RNA (9.8%) compared to juvenile camels (16.1% and 1.7% respectively, *p*-value < 0.0001). Comparable investigations somewhere else likewise showed a higher seroprevalence in adult than in juvenile camels [24]. Our results showed that the overall seropositivity of camels in Tunisia was 87.3%. A previous study showed lower MERS-CoV seropositivity in adult (54%) and juvenile (30%) camels raised for meat production in Tunisia [15]. Our study showed 0.7% detection rate of MERS-CoV in nasal swabs of camels in Tunisia. The detection of MERS-CoV has been described in East African countries but none from Uganda. The results of this study showed that camels in Uganda have a rate of 66% of MERS-CoV antibodies. In Uganda, camels reside in eastern regions that share cultural and geographical characteristics with Kenya [25]. A previous study showed that camels in Kenya have a high seroprevalence rate of antibodies for MERS-CoV [26].

This was the first study conducted for surveillance of MERS-CoV in dromedary camels in Senegal. The dromedary camels in Senegal have 65.1% seroprevalence rate of antibody to MERS-CoV and 5.5% detection rate of genetic material of MERS-CoV suggesting that the virus is widespread in Senegal. In Iraq, camels reside in only four provinces: Anbar, Najaf, Basrah, and Muthanna. The overall seroprevalence of MERS-CoV in Iraq was 43.7%. In this study, serum samples were collected from only two provinces: Najaf and Muthanna. There was no significant difference in seropositivity to MERS-CoV between the two provinces in Iraq. There was no significant difference in seropositivity of MERS-CoV in serum samples collected from different regions in KSA. Previous studies carried out in KSA revealed regional difference in seropositivity to MERS-CoV ranged from 100% to 37% [27]. The virus detection rate in camels was detected in 3.1%. The main source of imported camels in KSA is Somalia and Sudan through the KSA Red Sea ports of Jizan and Jeddah.

Phylogenetic analysis of partial spike nucleotide sequences containing RBD showed that MERS-CoV in camels in Egypt clusters together with African MERS-CoV in a clade C. Previous studies found that MERS-CoV circulating in dromedaries in Egypt, Burkina Faso, Nigeria, Morocco, and Ethiopia are genetically distinct from viruses in the Arabian Peninsula [28,29]. Interestingly, the obtained sequences from infected camels with MERS-CoV in KSA were distinct from previously detected MERS-CoV in Arabian Peninsula and closely related to clade C of the African detected viruses. The detected viruses were from Al-Khomra near Jeddah. Most of camels in this region were imported from Sudan and Somalia suggesting the role of imported camels in the emerging of this clade into KSA. In contrast to data from a previous study [28], the three obtained sequences from Senegal in West Africa were peculiarly related to cluster B circulating in Asia. It is important to note that those findings may have been affected by the relatively short sequenced fragment and full genome sequences of these samples will be more realistic.

Our longitudinal study on three dromedary camels’ herds showed high seroprevalence in adult camels in agreement with previous findings from different countries [24]. All juvenile camels in the Matrouh herd showed low antibody titer. Two Egyptian herds had evidence of MERS-CoV infection in state of the presence of MERS-CoV neutralizing antibodies of some of infected camels. In contrast, a study of infection within camel herds including adult and juvenile camels showed seronegative juvenile camels exposed to viral infection more than seropositive adult animals [30]. A limitation of longitudinal study in Esna herd is the short period within which the sampling was carried out. No virus was detected in Jordan herd. Comparing the antibody titers among three herds, Jordan herd showed the highest seroprevalence titers without any significant drop in neutralizing antibodies. Data from this study showed that camels from Matrouh herd that were infected did not demonstrate a significant increase in titers. The source of infection is not clear. Another study limitation concerning longitudinal surveillance is the fact that the effect of seasonal climatic variation and difficulty to compare RNA activity between different sites impaired our capacity to draw uniform conclusions from all the sites. Another study limitation, urine, milk, and feces samples were not included in our study that give insights into MERS-CoV replication and virus shedding.

In conclusion, the very high prevalence of MERS-CoV neutralizing antibodies in tested camels in different countries with a larger geographical range indicates the widespread nature of the virus in camels. A systematic active surveillance and longitudinal studies for MERS-CoV are needed to understand the epidemiology of the disease and dynamics of infection. The very high seroprevalence detected in camels warrants the initiation of an active surveillance study for MERS CoV in humans, particularly those that are at higher risks of exposure to MERS-CoV infections such as camel traders and abattoir workers.

## Figures and Tables

**Figure 1 viruses-11-00717-f001:**
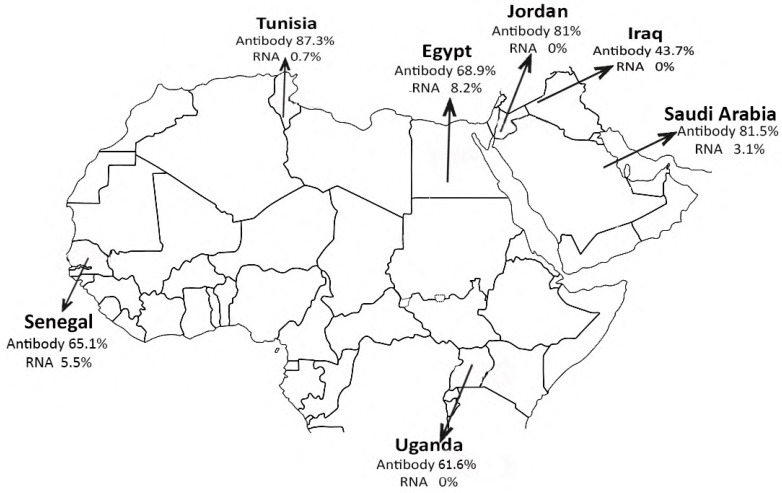
Countries involved in this study. Results are indicated for each country as percentage of detected virus (RNA) and seropositive rate (Antibody) for Middle East respiratory syndrome coronavirus (MERS-CoV). Maps adapted from http://d-maps.com/index.php.

**Figure 2 viruses-11-00717-f002:**
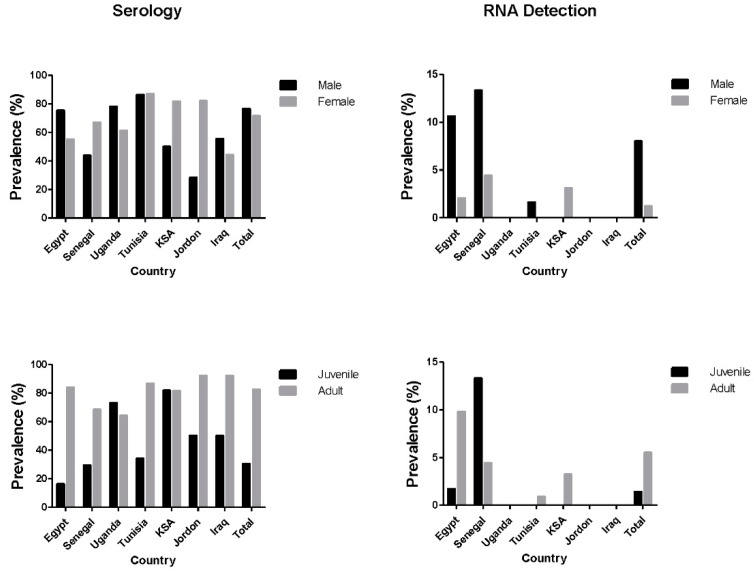
Comparison of the prevalence rate of MERS-CoV antibodies in serum (left) and RNA detection in nasal swabs (right) collected from camels in seven countries under investigation by age and sex.

**Figure 3 viruses-11-00717-f003:**
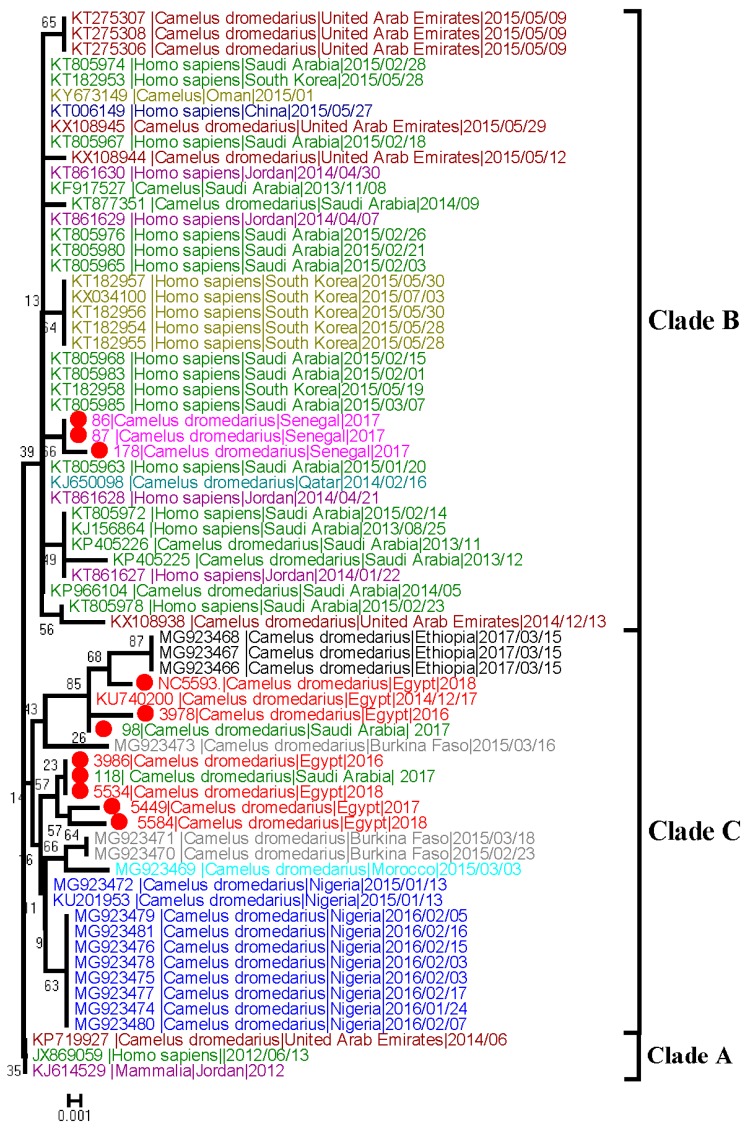
Phylogenetic tree of the partial Spike gene of RBD of MERS-CoV (=701 bp). Tree was generated using MEGA6 with bootstrap method and Kimura 2-parameter model. Sequences obtained in this study are labelled with a red circle, other MERS-CoV sequences from different countries are categorized by colors.

**Figure 4 viruses-11-00717-f004:**
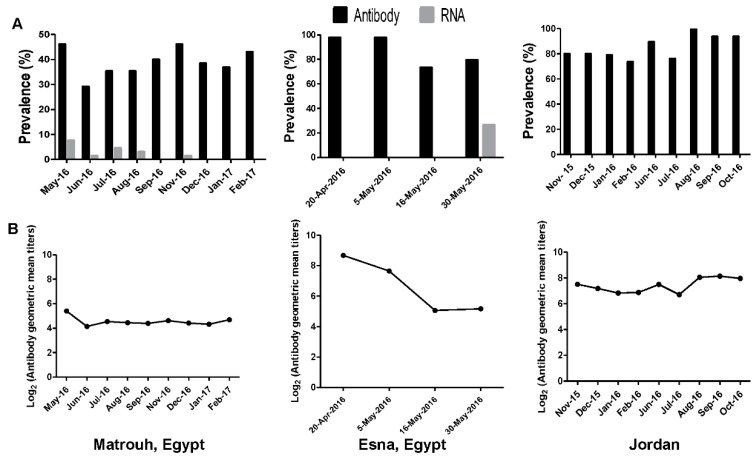
Longitudinal study to follow up the prevalence of MERS-CoV in two herds of camels in Egypt (Matrouh and Esna) and a herd in Jordan. (**A**) Indicates the seropositive rate of MERS-CoV in three herds and percentage of confirmed MERS-CoV RNA by RT-PCR. (**B**) indicates the geometric mean of antibody titers.

**Table 1 viruses-11-00717-t001:** MERS-CoV microneutralization and nasal swab RT-PCR test results by sampling site, age, sex and animal origin.

Country	Sampling Variable	Microneutralization Assay		Nasal Swabs RT-PCR	
		No. Tested	No. (Positive)	No. Tested	No. (Positive)
**Egypt**	***Sampling Site***				
	Live Animal Market	309	254 (82.2)	286	12 (4.1)
	Free Herd	187	129 (68.9)	187	3 (1.6)
	Farm	924	474 (51.2)	944	33 (3.4)
	Quarantine	164	143 (87.1)	164	2 (1.2)
	Slaughterhouse	449	401 (89.3)	649	133 (20.4)
	*p*-value	<0.0001		<0.0001	
	***Governorate***				
	Aswan	340	300 (88.2)	317	4 (1.2)
	Beheira	68	39 (57.3)	68	3 (4.4)
	Cairo	202	178 (88.1)	265	57 (21.5)
	Giza	287	236 (82.2)	430	74 (17.2)
	Qaliobia	93	87 (93.5)	96	13 (13.5)
	Luxor	238	197 (82.7)	236	16 (6.7)
	Matrouh	715	297 (41.5)	728	16 (2.1)
	Sharkia	90	67 (74.4)	90	0 (0)
	*p*-value	<0.0001		<0.0001	
	***Age***				
	Juvenile	447	72 (16.1)	462	8 (1.7)
	Adult	1586	1329 (84)	1768	175 (9.8)
	*p*-value	<0.0001		<0.0001	
	***Sex***				
	Male	1400	1053 (75.4)	1592	169 (10.6)
	Female	633	348 (55)	638	14 (2.05)
	*p*-value	<0.0001		<0.0001	
	Animal Origin				
	Imported	922	798 (86.5)	1099	147 (13.3)
	Local	1111	603 (54.2)	1131	36 (3.1)
	*p*-value	<0.0001		<0.0001	
	Total	2033	1401 (68.9)	2230	183 (8.2)
**Senegal**	***Sampling Site***				
	Gandon	28	18 (64.2)	0	0
	Ndaye	23	23 (100)	23	1 (4.3)
	Gantour	20	19 (95)	20	1 (5)
	Ndoye	43	30 (69.7)	16	3 (18.7)
	Tongon	13	3 (23.07)	13	0 (0)
	Toug	37	36 (97.2)	21	0 (0)
	Rao	34	0	34	2 (5.8)
	*p*-value	<0.0001			ns
	***Age***				
	Juvenile	17	5 (29.4)	15	2 (13.3)
	Adult	181	124 (68.5)	112	5 (4.4)
	*p*-value	<0.01		ns	
	***Sex***				
	Male	16	7 (43.7)	15	2 (13.3)
	Female	182	122 (67)	112	5 (4.4)
	*p*-value	ns		ns	
	Total	198	129 (65.1)	127	7 (5.5)
**Uganda**	***Sampling Site***				
	Moroto	443	271 (61.1)	443	0
	Amudat	57	37 (64.9)	57	0
	*p*-value	ns		ns	
	***Age***				
	Juvenile	150	78 (52)	150	0
	Adult	350	230 (65.7)	350	0
	*p*-value	<0.003		ns	
	***Sex***				
	Male	145	89(61.3)	145	0
	Female	355	219 (61.6)	355	0
	*p*-value	ns		ns	
	Total	500	308 (61.6)	500	0 (0)
**Tunisia**	***Sampling Site***				
	Douz	293	228 (77.8)	341	0
	Ghlissia	34	31 (91.1)	34	1 (2.9)
	Gollaa	24	13(54.1)	24	0
	Ksar Ghilane	58	38 (65.5)	59	8 (13.5)
	Ghidma	208	208 (100)	338	0 (0)
	Jamnah	41	41 (100)	45	0 (0)
	Zaafrana	124	124 (100)	229	0 (0)
	*p*-value	<0.0001		<0.0001	
	***Age***				
	Juvenile	28	28 (100)	82	0 (0)
	Adult	754	655 (86.8)	988	9 (0.9)
	*p*-value	0.02		ns	
	***Sex***				
	Male	281	243 (86.4)	550	9 (1.6)
	Female	‘	440	620	0
	*p*-value	ns		0.001	
	Total	782	683 (87.3)	1170	9 (0.7)
**KSA**	***Sampling Site***				
	Alkhomra	36	29 (80.5)	36	1 (2.7)
	alsheaeba	17	10 (58.8)	17	0 (0)
	Asfan	62	56 (90.3)	62	0 (0)
	Dahban	26	19 (73.07)	26	3 (11.5)
	Khlees	10	8 (80)	11	0 (0)
	Mecca	19	17 (89.4)	19	0 (0)
	Umm al Jurm	12	10 (83.3)	13	0 (0)
	Alad	24	22 (91.6)	24	3 (12.5)
	Um Marekh	16	15 (93.7)	16	0 (0)
	*p*-value	ns		0.03	
	***Age***				
	Juvenile	11	9 (81.8)	11	0 (0)
	Adult	211	172 (81.5)	213	7 (3.2)
	*p*-value	ns		ns	
	***Sex***				
	Male	2	1 (50)	2	0 (0)
	Female	220	180 (81.8)	222	7 (3.1)
	*p*-value	ns		ns	
	Total	222	181 (81.5)	224	7 (3.1)
**Jordan**	***Sampling Site***				
	Aqaba	304	246 (80.9)	304	0 (0)
	***Age***				
	Juvenile	82	41 (50)	82	0 (0)
	Adult	222	205 (92)	222	0 (0)
	*p*-value	<0.0001			
	***Sex***				
	Male	7	2 (28.5)	7	0 (0)
	Female	297	244 (82.1)	297	0 (0)
	*p*-value	<0.003			
	Total	304	246 (81)	304	0 0 (0)
**Iraq**	***Sampling Site***				
	Wasit	14	13 (92.8)	13	0 (0)
	Muthana	13	1 (7.6)	13	0 (0)
	*p*-value				
	***Age***				
	Juvenile	6	2 (33.3)	6	0 (0)
	Adult	21	12 (57.1)	20	0 (0)
	*p*-value	ns			
	***Sex***				
	Male	9	5 (55.5)	8	0 (0)
	Female	18	8 (44.4)	18	0 (0)
	*p*-value	ns			
	Total	32	14 (43.7)	26	0 (0)

## Data Availability

The raw data supporting the conclusions of this manuscript has been submitted to GenBank and will be made available by the authors, without undue reservation, to any qualified researcher.

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
