# Peer review of "Middle East Respiratory Syndrome Coronavirus (MERS-CoV) in Dromedary Camels in Africa and Middle East"

_viruses, 2019, doi:10.3390/v11080717_

Round 1

Reviewer 1 Report

The paper is a generally well-written addition to the story of camels, MERS-CoV and MERS. It is disappointing that the very obvious opportunity to sample urine, milk and faeces for the presence of viruses was not taken in this large camel study. As the authors rather quietly and correctly point out (ln83), there are risks attached to the handling and consumption of these camel products that have yet to be suitably studied. This oversight does not impact on the quality of the content presented, however, although its omission as a limitation in the discussion is glaring.

ln40. "and a phylogenetic tree..."

ln52 "..widespread distribution of.."?

ln52-53. Please re-word "A systematic active surveillance" to be more descriptive. It currently is not clear what this phrase would entail. Also repeated at ln347. Also, is this all for use "in humans". Please clarify.

ln75. Was it lack of surveillance, or absence of knowledge of the virus until relatively recently? Please clarify.

ln76. "disease acquired" would be better described as identified or reported.

ln81. "...use of personal protective.." 

ln342. What is the seasonal effect the authors hint at? Please elaborate on this.

ln346. "..widespread nature of the virus..."

Author Response

Review Report Form (1)

Open Review

English language and style

( ) Extensive editing of English language and style required
( ) Moderate English changes required
(x) English language and style are fine/minor spell check required
( ) I don't feel qualified to judge about the English language and style

Yes

Can be improved

Must be improved

Not applicable

Does the introduction provide sufficient background and include all relevant references?

(x)

( )

( )

( )

Is the research design appropriate?

(x)

( )

( )

( )

Are the methods adequately described?

(x)

( )

( )

( )

Are the results clearly presented?

(x)

( )

( )

( )

Are the conclusions supported by the results?

(x)

( )

( )

( )

Comments and Suggestions for Authors

The paper is a generally well-written addition to the story of camels, MERS-CoV and MERS.

It is disappointing that the very obvious opportunity to sample urine, milk and faeces for the presence of viruses was not taken in this large camel study. As the authors rather quietly and correctly point out (ln83), there are risks attached to the handling and consumption of these camel products that have yet to be suitably studied. This oversight does not impact on the quality of the content presented, however, although its omission as a limitation in the discussion is glaring

Added in the limitation section.

ln40. "and a phylogenetic tree..."

Corrected

ln52 "..widespread distribution of.."?

Modified

ln52-53. Please re-word "A systematic active surveillance" to be more descriptive. It currently is not clear what this phrase would entail. Also repeated at ln347. Also, is this all for use "in humans". Please clarify.

Clarified

ln75. Was it lack of surveillance, or absence of knowledge of the virus until relatively recently? Please clarify. ln76. "disease acquired" would be better described as identified or reported.

Modified

ln81. "...use of personal protective.." 

Changed

ln342. What is the seasonal effect the authors hint at? Please elaborate on this.

Changed

ln346. "..widespread nature of the virus..."

Changed

Submission Date

09 July 2019

Date of this review

25 Jul 2019 04:41:07

Reviewer 2 Report

This submission provides valuable information on the prevalence of MERS-CoV in dromedary camels.  A positive feature of the submission is with its thorough analyses, notably the evaluation of nearly 8000 camel nasal swabs and serum samples for MERS-CoV RNA and neutralizing antibodies.  The data are presented adequately, although statistical analyses should be included in figures and tables if possible.  The conclusions are, for the most part, supported by the findings.  

Specific comments:

Include results of statistical analyses in the figures and tables, if possible.

Include additional description of the phylogenetic tree in Fig. 3.  From what length of sequenced fragment was the tree constructed?   What was the extent of polymorphism in the sequenced fragment?  Is the tree distinct from those trees constructed on the basis of polymorphisms in the ORF1A or N genes that were PCR amplified and sequenced?  It was noted on line 330 that previous phylogenetic trees may have been affected because they were based on short sequenced fragments.  What makes the tree in Fig. 3 better?

The authors suggest that virus present in seropositive animals is coming from reinfections.   If so, then the reinfecting viruses may be variants, for example, antibody escape variants.  Do the sequences of the presumed reinfecting viruses separate into a separate group (ie., clade) on the basis of phylogenetic analyses?  

Amongst the most interesting statements in the paper is on line 76, "there is no zoonotic MERS disease acquired in Africa".   The results in the paper show high seroprevalence as well as recoverable MERS-CoV RNA in African samples.  In my opinion, the authors are entitled to speculate (in the discussion) on this conundrum posed by high virus loads in Africa without zoonotic transmission.  

Several grammatical corrections must be made to improve the accuracy and flow of the text.

Author Response

Review Report Form (2)

Open Review

English language and style

( ) Extensive editing of English language and style required
(x) Moderate English changes required
( ) English language and style are fine/minor spell check required
( ) I don't feel qualified to judge about the English language and style

Yes

Can be improved

Must be improved

Not applicable

Does the introduction provide sufficient background and include all relevant references?

(x)

( )

( )

( )

Is the research design appropriate?

( )

(x)

( )

( )

Are the methods adequately described?

( )

(x)

( )

( )

Are the results clearly presented?

(x)

( )

( )

( )

Are the conclusions supported by the results?

( )

(x)

( )

( )

Comments and Suggestions for Authors

This submission provides valuable information on the prevalence of MERS-CoV in dromedary camels.  A positive feature of the submission is with its thorough analyses, notably the evaluation of nearly 8000 camel nasal swabs and serum samples for MERS-CoV RNA and neutralizing antibodies.  The data are presented adequately, although statistical analyses should be included in figures and tables if possible.  The conclusions are, for the most part, supported by the findings.  

Specific comments:

Include results of statistical analyses in the figures and tables, if possible.

Include additional description of the phylogenetic tree in Fig. 3. From what length of sequenced fragment was the tree constructed?   What was the extent of polymorphism in the sequenced fragment?  Is the tree distinct from those trees constructed on the basis of polymorphisms in the ORF1A or N genes that were PCR amplified and sequenced? It was noted on line 330 that previous phylogenetic trees may have been affected because they were based on short sequenced fragments.  What makes the tree in Fig. 3 better?

The length of amplicons and rational of the tree was added in the results and figure ligand as suggested

The authors suggest that virus present in seropositive animals is coming from reinfections.   If so, then the reinfecting viruses may be variants, for example, antibody escape variants.  Do the sequences of the presumed reinfecting viruses separate into a separate group (ie., clade) on the basis of phylogenetic analyses?

  Our group and other showed the reinfections of seropositive camels are common.  

Amongst the most interesting statements in the paper is on line 76, "there is no zoonotic MERS disease acquired in Africa".   The results in the paper show high seroprevalence as well as recoverable MERS-CoV RNA in African samples.  In my opinion, the authors are entitled to speculate (in the discussion) on this conundrum posed by high virus loads in Africa without zoonotic transmission.  

Added in the discussion

Several grammatical corrections must be made to improve the accuracy and flow of the text.